# Analysis of Innovative Methods' Effectiveness in Teaching Foreign Languages for Special Purposes Used for the Formation of Future Specialists' Professional Competencies

**Irina G. Belyaeva [1], Ekaterina A. Samorodova [2,\*], Olga V. Voron [3] and Elena S. Zakirova [4]**

[1]  Department of German language, Moscow State Institute of International Relations (MGIMO University),
    76 Pr. Vernadskogo, Moskow 119454, Russia
[2]  Department of French language, Moscow State Institute of International Relations (MGIMO University),
    76 Pr. Vernadskogo, Moskow 119454, Russia
[3]  Institute for Social Policy, Higher School of Economics—National Research University,
    20 Myasnitskaya Ulitsa, Moscow 101000, Russia
[4]  Department of Foreign Languages, Peoples' Friendship University of Russia, 6 str. Mikluho Maklaya,
    Moskow 117198, Russia
\*  Correspondence: samorodova.ekaterina.78@mail.ru

**Abstract:** One of the most important tasks of higher educational institutions is the training of specialists to be able to adapt to changes in their professional life. At the end of the twentieth and the beginning of the 21st centuries, some methods for developing foreign language competence, needed for their future professional activity, were created by teachers. However, the effectiveness of these methods has not been studied. This fact has aroused the authors' interest and generated the idea about the necessity to conduct scientific research in order to identify the most effective methods of teaching foreign languages for special purposes. Methods: The given research paper is based on the analysis of Russian and foreign scholars' scientific works covering the problem of teaching foreign languages for special purposes to the students of humanitarian professions, as well as on the basis of the results from questioning students of bachelor degree programs who study foreign languages for special purposes in the field of humanitarian professions, and also of the results from questioning teachers specializing in teaching foreign languages for special purposes. Results: In the students' opinion, the most effective methods of teaching foreign languages for special purposes in the field of humanitarian professions are the following: discussion, ICT (information and communication technologies), and SCRUM (framework that helps teams work together, encourages team to learn through on a problem). According to the interviewed teachers' opinion, the most effective methods are discussion, ICT, and round table. The "dilemma" method is the least effective according to the students. As for the teachers, the less effective method is CLIL (content and language integrated learning). Conclusions: The study showed some common views among teachers and students concerning the effectiveness of methods of teaching foreign languages for professional purposes, such as discussion and ICT. The effectiveness of the discussion method is explained by the fact that it allows the integration of students' knowledge from different areas when solving a problem and provides an opportunity to apply language knowledge and skills into practice. This contributes to forming students' ability to think clearly, to perceive information critically, to highlight the main idea and find the means and arguments to confirm and substantiate it, and, consequently, to improve the understanding of any theoretical material. The use of ICT in the educational process allows the efficiency of the educational process itself to be improved significantly and leads to new approaches and organizational forms of educational work. In fact, while preparing educational programs and creating didactic materials, special attention should be given to the implementation of ICT methods and discussions in educational activities. Nevertheless, the respondents' subjective opinion should

not reduce the scientific value and effectiveness of other methods of teaching a foreign language for professional purposes. The authors of the paper believe that methods that have not found much support from students and teachers should be studied more thoroughly and carefully. To this end, it could be recommended to organize special training seminars that would allow teachers to be informed of new methods of teaching foreign languages for professional purposes, of their particularities, and to help their active implementation in the learning process.

**Keywords:** foreign language for special purposes; method of teaching foreign languages for special purposes; the effectiveness of teaching methods

---

## 1. Introduction

Modernity, as a complex philosophical concept, has no clear boundaries and is quite relative. Our contemporaries are united by the change of the state system that occurred in the nineties, which caused the collapse of the old ideals and values, left an imprint on people's minds, and changed their way of thinking and life. Moreover, technical progress that has rapidly been changing the world has brought nations together and has greatly influenced people's professional interaction. Professional-oriented learning of foreign languages does not stand aside but instead responds vividly to all the changes taking place in modern society. Communication in the professional sphere of Soviet citizens, with specialists from other countries, was minimal. Therefore, teaching foreign languages for special purposes was primarily aimed ar developing the skills of reading and translating the special texts mainly through reproductive methods. With the fall of the so called "iron curtain", the need for specialists having oral communication skills has considerably grown. In this regard, new requirements for methods of teaching foreign languages for special purposes necessitated the development of students' listening and speaking skills. Changes in the tasks of professional education coincided with the establishment of the Institute of Productive Learning in Leningrad in 1992. This made it possible to develop new competences on the basis of the ideas brought into pedagogy by a team of Leningrad teachers, whose opinions were shared by their foreign colleagues. The result of this research institute activity has become the developed methods, such as cognitive, creative, and organizational methods, which have been used for 30 years to varying degrees in all Russian higher educational institutions. Such methods are fundamentally different from the reproductive ones that prevailed over several centuries and are considered in this article as modern. The main task of modern higher education is to train specialists to be able to not only master the foreign language for special purposes but also to be able to adapt to all changes in their professional life caused by the rapid progress and development of social thought. These factors allow students to develop their self-education ability, allowing them to find new knowledge [1] (p.4). Professionally-oriented teaching of foreign languages aims at providing graduates with the ability to use foreign languages in professional communication. Therefore, being of practical usage, the study of foreign languages for special purposes aims at solving the communicative problems in students' future professional activity. We revealed that some scientific papers dedicated to the study of foreign language for special purposes (LSP) teaching methods provide descriptions of their main characteristics and proved their effectiveness from the authors' subjective point of view [2,3]. Many articles on the problem of LSP teaching methods' effectiveness highlight only one of the methods, such as the interval repetition method "spaced repetition technique" [4–6] or individual information and communication technologies (ICT) techniques [7–10]. We suppose that a good theoretical study of a scientific problem does not always mean its highly effective implementation in practice. The majority of educationalists have created effective methods of LSP teaching methods in the past centuries and, at the present time, have been offering various forms for their active usage to develop foreign language competence that is needed in students' future professional activity. The analysis of scientific literature has shown that scientific works reflecting such data have not been developed yet. When improving

teachers' skills, many educators master new methods of foreign language teaching that theoretically meet the requirements of future specialists. However, the problem of dealing with the usage of theoretical knowledge in practice has not been solved yet. Moreover, the effectiveness of the LSP teaching methods aiding specialists to be successfully competitive in real professional activity, taking the modern labor market conditions into consideration, has not been studied.

The main objective of this paper was to identify, through research, the most effective LSP teaching methods allowing students to perfect special sublanguage and take part in professional communication.

In this paper, the authors have solved the following research problems:

(1) identifying in practice the methods of teaching foreign language for special purposes in humanitarian professions;
(2) assessing the effectiveness of LSP teaching methods put into practice by teachers; and
(3) assessing the effectiveness of the methods taking humanitarian faculties students' opinion into consideration.

## 2. Materials and Methods

The given scientific research involves the analysis of theoretical material concerning the methods of teaching foreign languages for special purposes and their effectiveness. The results of scientific literature analysis have shown that the most widely used methods in teaching foreign language in modern educational practice can be identified as: the method of project [11,12], case studies [13,14], ICT [15–19], brainstorming [20,21], the role-playing method [22,23], tandem [24,25], extensive reading [26], the method of podcasts [27,28], the associative method [29,30] training in collaboration or cooperative learning, collaborative learning [31,32], sliding [33], the method of contrastive linguistics [34], discussions [35], dilemma [36], jigsaw reading [37], the method of theatre production [38], SCRUM [39], round table [40], peer review [41,42] mnemonics [43,44], the grammar–translation method [45], the direct method [45], the method of reading [45], the audio–lingual method [46], Dr. West's flipped learning or lipped classroom [47,48]; content and language integrated learning (CLIL) [49,50]; and the cooperative learning method [51,52].

At the second stage of the study, the authors considered productive methods of teaching foreign language for special purposes used in higher educational system in Russia. At the next stage, these methods were offered to the teachers and students for them to select and assess the most effective ones.

To identify the degree of effectiveness of a particular method of LSP teaching in a nonlinguistic university (faculty), a written questionnaire was completed by teachers and students.

The written teacher questionnaire was carried out in order to identify modern methods of teaching foreign languages in special purposes to the students of humanitarian specialties. The questionnaire involved the selection of methods based on the individual judgment of the participant, based on their personal experience in the field of teaching a foreign language for special purposes.

In the first part of the questionnaire, the teachers needed to name the university in which they teach the language for special purposes at the present time, their age, and gender. In the second part of the questionnaire, it was proposed to consider the modern methods of teaching a foreign language in the table and choose the ones used by them in pedagogical practice ticking one of the following (Table 1):

**Table 1.** Methods of teaching foreign languages for special purposes to students of humanities.

| Teaching Methods | | | |
|---|---|---|---|
| Methods with Using ICT | Learning in Collaboration | Methods of Problem | Methods of Play |
| | Method of Project | Discussion | Role-play Method |
| | Jigsaw Reading | Dilemma | Business Play |
| | SCRUM | Case Study | Educational Firm |
| | Cooperative Learning, Collaborative Learning | Round Table | Tandem |
| | | Brainstorming | Method of Mnemonics |
| | | Peer Review | Extensive Reading |
| | | | Flipped Learning/Flipped Classroom |
| | | | Role-play Method |
| | | | CLIL |

Respondents were invited to select any number of methods. In the third part of the questionnaire, teachers were asked to indicate the additional used modern methods of teaching LSP which are not listed in the given list.

The students were asked to fill out a questionnaire to assess the effectiveness of modern methods of LSP teaching to humanitarian specialties. The survey in the form of a questionnaire was conducted in order to determine the most effective methods of teaching LSP to students of humanities which are used in practice. The questionnaire included an assessment of the effectiveness of modern teaching methods, presented in Table 1, by assigning them the corresponding points. The evaluation of effectiveness was of an individual participant's judgment based on their personal experience in learning LSP by the students of humanitarian faculties.

In the first part of the questionnaire, it was necessary to indicate the name of the university and the direction (profile) of training, as well as the year of studying at the university (the course in which the student is studying), the acquired language level in this course (A1, A2, B1, B2, and C1), their age, and gender. In the second part of the questionnaire, it was proposed to consider the proposed methods of teaching LSP (Table 1) and to evaluate the effectiveness of each of them. The assessment was made on a scale from 5 points to 0 point, where:

5—method of a very high efficiency;
4—method of high efficiency;
3—method of average efficiency;
2—method of low efficiency;
1—method of very low efficiency;
0—an inefficient method.

Several methods could be evaluated with the same number of points. Methods that are not used in teaching LSP to students of humanities were not evaluated.

## 3. Results

To solve the first problem of the given research aimed at identifying the methods of teaching LSP in humanitarian specialties used in teaching practice, 35 teachers from five leading universities in Russia and one university in Germany were interviewed: MGIMO University, RANEPA, Kuban State Technologic University, RUDN University, Financial University, and University of Duisburg-Essen (Germany) (Figure 1):

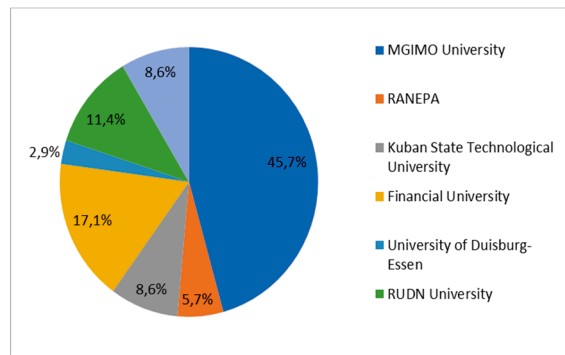

**Figure 1.** Distribution of teachers in educational institutions.

The age of the teachers was between 27 and 63 years old, of whom 91.4% were women and 8.6% were men who teach English, German, and French (Figure 2):

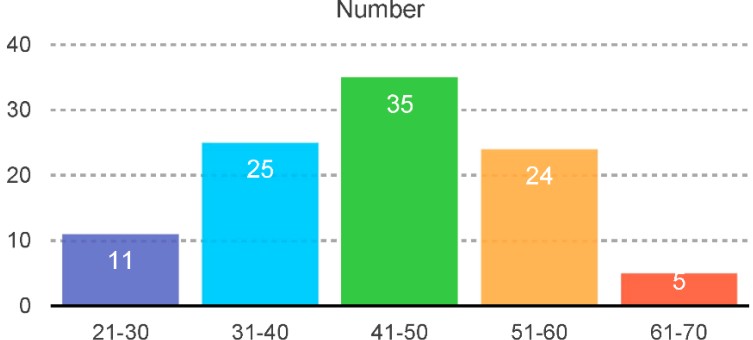

**Figure 2.** Distribution of teachers by age.

The results of the research showed that all the identified productive methods used in teaching LSP according to their popularity among teachers who participated in this research can be distributed as follows (Figure 3)

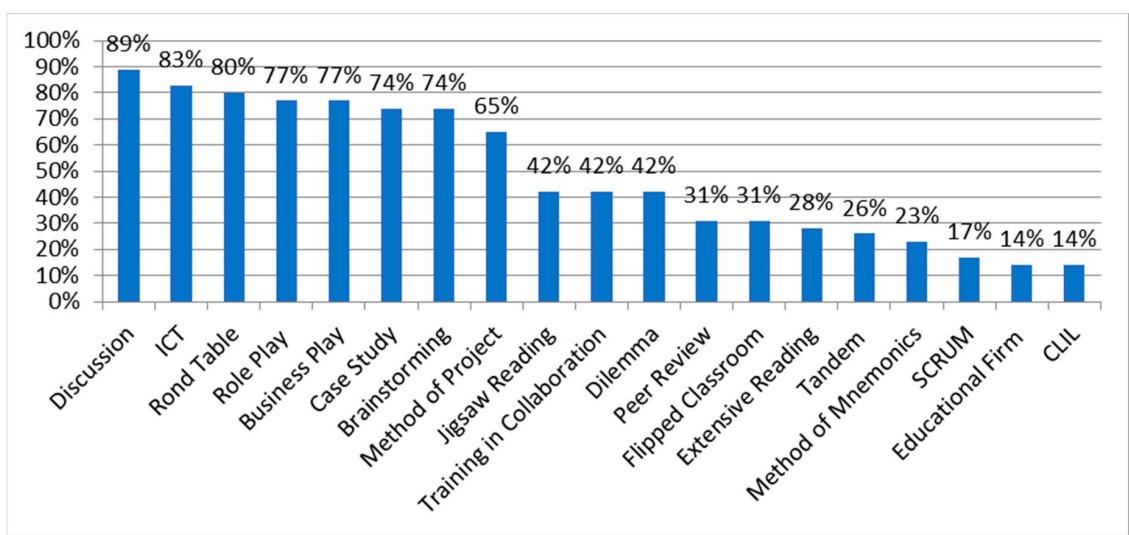

**Figure 3.** Distribution the methods by their popularity among teachers of foreign language for special purposes (%).

When analyzing Table 2, a significant variation in the frequency of using methods of LSP teaching should be noted. The most frequently used method is discussion, having a very high degree of

application: It was pointed out by 89% of teachers. At the same time, such methods as educational firm and CLIL are applied in practice only by 14% of the teachers: They are 6 times less likely to be used than the method of discussion. The substantial predominance of some methods used in LSP teaching over other methods can be explained in the teachers' opinion by the factors concerning (1) the degree of their effectiveness, (2) the various levels of teachers' proficiency and their ability to use them, and (3) the teachers' experience and practice.

**Table 2.** Assessment of the effectiveness of modern methods of teaching foreign language for humanitarian specialties from the teacher's standpoints, where 5 is the most effective method and 0 is not an effective method.

| Teaching Methods | 1. 21–31 | 2. 31–40 | 3. 41–50 | 4. 51–60 | 5. 61–70 |
|---|---|---|---|---|---|
| Method of Project | 2 | 3 | 3 | 5 | 2 |
| Cooperative Learning, Collaborative Learning | 2 | 2 | 2 | 2 | 2 |
| Discussion | 5 | 5 | 5 | 5 | 5 |
| Role-play Method | 3 | 5 | 5 | 5 | 1 |
| Dilemma | 2 | 2 | 2 | 2 | 2 |
| SCRUM | 0 | 1 | 1 | 0 | 0 |
| Round Table | 5 | 4 | 4 | 5 | 5 |
| Brainstorming | 3 | 3 | 5 | 3 | 3 |
| Peer Review | 2 | 2 | 2 | 1 | 1 |
| Extensive Reading | 0 | 2 | 2 | 1 | 2 |
| Flipped Learning/Flipped Classroom | 2 | 2 | 2 | 2 | 0 |
| CLIL | 0 | 1 | 1 | 0 | 0 |
| Methods with using ICT | 5 | 5 | 5 | 5 | 4 |
| Business Play | 3 | 4 | 4 | 2 | 5 |
| Educational Firm | 0 | 1 | 1 | 0 | 0 |
| Case Study | 3 | 3 | 3 | 3 | 5 |
| Jigsaw Reading | 2 | 2 | 2 | 2 | 2 |
| Tandem | 0 | 1 | 2 | 2 | 0 |
| Method of Mnemonics | 0 | 2 | 1 | 1 | 0 |

The first positions are occupied by methods that contain the most expressed professional component. These methods are discussion and round table. The discussion and the round table as teaching methods are the simplest from the point of view of their organization and implementation at the classes, which allow discussing various aspects of future professional activity in a foreign language. At the same time, these types of educational activities do not require an exact answer to the question posed or an exact solution of the problem. They allow the students to determine and evaluate if the answer to the question is correct or not based on their special knowledge obtained in lectures and seminars on subjects related to their future profession. The methods of discussion and round table differ from the case study method, which requires a specific solution to a specific professional task. The usage of the case study method at a foreign language lesson requires from the teacher not only professional mastering of a foreign language but also deep knowledge of the specialty in which their students are taught. This helps the teacher to formulate and solve professionally-oriented tasks in the lesson. The lack of deep knowledge in the specialty, the language of which teachers–linguists should teach the students, is the main reason of teacher's inability to solve, for example, legal or economic problems as one of the tasks in the case study method. This tendency is also confirmed by

the fact that the CLIL method occupies the last position in the ranking of the LSP teaching methods' popularity among teachers, since use of this method means a brilliant mastery of the special subject. The training of foreign language teachers in linguistic and pedagogical universities is not aimed at the training of specialists who are fully proficient in, for example, foreign language used in such fields as jurisprudence or economics. The only aspect that can be taught quite professionally in nonlinguistic universities, for example, at political science or international relations faculties, is political translation, since future foreign language teachers study it. The leading methods used by LSP teachers in higher educational institutions are those using information and communication technologies. These methods allow the teacher to apply a wide range of activities in LSP teaching process: from working on relevant articles on the specialty to creating educational firms and the application of the tele-tandem method. The results of the research showed that all the identified productive methods used in teaching LSP according to their efficiency among teachers of different ages who participated in this research can be distributed as follows (Table 2).

To solve the second problem of our research, aimed at evaluating the effectiveness of the methods used in the practice of LSP teaching to students of humanities, the authors asked students mastering a professionally-oriented foreign language for humanitarian professions to fill the questionnaire and to point at the most effective methods from their standpoint. At this stage of our research, the participants were 67 students from the 1st, the 2nd, the 3rd, the 4th years of studying on bachelor's degree programs at MGIMO University, Moscow Polytechnic University, Financial University, and Russian University of Peoples' Friendship. They were students studying English, German, French, and Chinese LSP. They were 68.7% female and 31.3% male at ages between 17 to 24 years old, and they had reached A1, A2, B1, B2, and C1 in their study levels. When determining the effectiveness of the methods used in teaching a foreign language for special purposes, we considered the students' position to be very important, as they receive a sufficient understanding of the requirements for them as future specialists. The results of the research are shown in Figure 4:

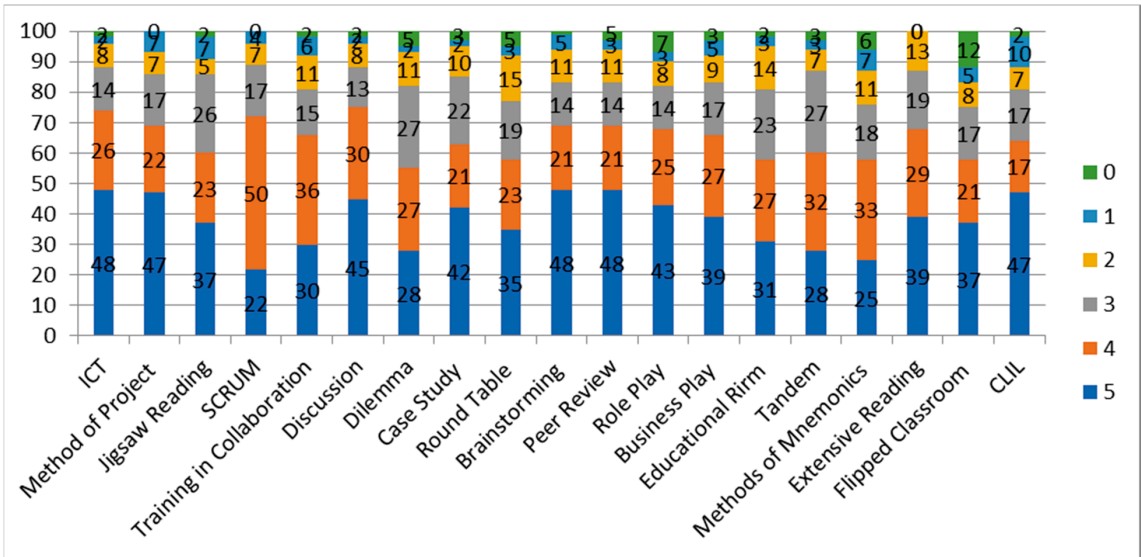

**Figure 4.** Assessment of the effectiveness of modern methods of teaching foreign language for humanitarian specialties (as a percentage) from the students' standpoints, where 5 is the most effective method and 0 is not an effective method.

In order to identify the most effective methods of teaching foreign languages for special purposes, the authors firstly collected the methods which were evaluated at "4" and "5" points by the students. The summarized points in percentage are shown in descending order in Figure 5:

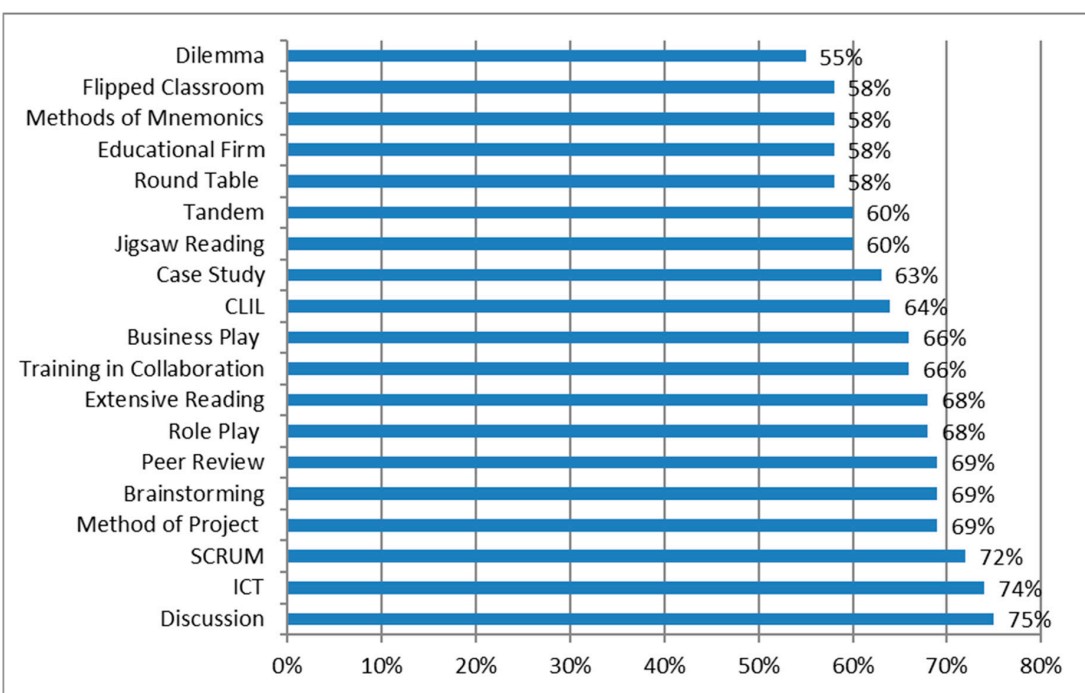

**Figure 5.** The most effective methods of teaching foreign languages for special purposes.

The evaluation of the methods given in the table has a relatively small variation. The students taking part in the research (55 to 75 percent of them) believe that the methods offered to them by the teachers of the foreign language for special purposes are effective. At the same time, it should be noted that none of the methods have been scored by the students less than 50%. This testifies to the fact that all the methods considered in the given research should be used in teaching foreign language for special purposes.

The top three most effective methods are discussion, ICT, and SCRUM. The least effective are such methods as dilemma.

## 4. Discussion

The majority of teachers (more than 50%) use such methods as discussion, ICT, round table, role-play, business play, case study, brainstorming, and the method of project at classes on foreign language for special purposes. The first three most popular methods pointed out by teachers include discussion, ICT, and round table. Moreover, it should be noted that students mentioned discussion and ICT as the most effective ones. This indicates that these methods, on the one hand, contain an orientation to the professional component of future specialists. In this regard, they are marked by students as the most effective. On the other hand, this result confirms the qualified usage of these methods by foreign language teachers. For example, the case study and training company methods train students for professional activities in a foreign language. The result of the research has shown that 74% of teachers use the method of case study in LSP teaching, but only 63% of the students called it effective. Despite the substantial professional orientation of the case-study method and its rather active use by LSP teachers, this situation can be explained by the insufficient level of teachers–linguists in special knowledge in the field of students' future profession. We consider that this fact significantly reduces the effectiveness of the case-study method. As for the "learning company" method, it is used by only 14% of teachers of foreign language for special purposes, despite its obvious high professional orientation. It was assessed as an effective method by 58% of students. These circumstances can be explained by the fact that there is no training program for teachers of a foreign language for special purposes in linguistic universities, and they have to get special knowledge in subjects related to the future profession of the students taught by them.

For the same reasons, not all professionally-oriented methods of teaching foreign languages for special purposes are actively used by teachers and are highly appreciated by students.

Therefore, methods that have not found much support from students and teachers should be more thoroughly and carefully studied.

To this end, it could be recommended to organize special training seminars that would allow teachers to be informed of new methods of teaching foreign languages for professional purposes, their particularities, and to help their active implementation in the learning process.

**Author Contributions:** I.G.B.—methodology development, concept study, initial text version preparation, data and proof collection, data analysis. E.A.S.—critical analysis and revision of the text, data and proof collection. O.V.V.—data and evidence collection, statistical processing, text correcting E.S.Z.—data and evidence collection, preparation of the English version of the article.

**Funding:** This research received no external funding.

**Acknowledgments:** This article is based on the results of the research conducted with the support of teachers of foreign languages departments of MGIMO University, as well as teachers from RANEPA, Kuban State Technologic University, and Financial University.

**Conflicts of Interest:** The authors declare no conflict of interest.

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
