# Peer review of "Analysis of Innovative Methods’ Effectiveness in Teaching Foreign Languages for Special Purposes Used for the Formation of Future Specialists’ Professional Competencies"

_education, doi:10.3390/educsci9030171_

Round 1

Reviewer 1 Report

It does not refer, in sufficient quantity and in an adequate manner, summarizing and commenting, as appropriate, on other research or work carried out in the field of the subject matter addressed. They should broaden the theoretical framework and support it with authority and relevant figures.

In terms of methodology, the criteria used for the selection of the sample are not presented, nor are the analysis criteria with which the information was approached accurately.

There are many shortcomings in the statistical analysis, which is limited to a few descriptive indications. The results should be expanded and detailed, discussed and reflected upon.

The discussion also lacks consistency. There is no certain link to the conclusions. There are certainly no recommendations that exhort the results obtained.

It is therefore necessary to improve these sections in order to give coherence and systematicity to the analyses and to the article as a whole.

Author Response

Dear colleague!

We warmly thank you for your huge work on our article, for your invaluable and highly qualified comments about its content. We are very pleased that you are interested in our research. We can say that this is already our common manuscript, since your contribution is invaluable. Let us submit our comments.

It does not refer, in sufficient quantity and in an adequate manner, summarizing and commenting, as appropriate, on other research or work carried out in the field of the subject matter addressed. They should broaden the theoretical framework and support it with authority and relevant figures.

Following your advice, we supplemented our research with new sources and references to the works of authoritative experts in this field. This enriched and improved our manuscript. Thank you for your comments. New sources in the article are highlighted in yellow.

In terms of methodology, the criteria used for the selection of the sample are not presented, nor are the analysis criteria with which the information was approached accurately.

In our study, we relied primarily on the criteria of the effectiveness of foreign language teaching methods for professional purposes according to the students' opinion. In this case, senior students acted as experts. We deeply believed that our research was very important for the lingo didactic because we were focused on the needs of students in the formation of professional skills. Few works have been written on this material, but the research itself is aimed at identifying the most effective methods for the purpose of their use in practice: the creation of textbooks, programs, and so on.

There are many shortcomings in the statistical analysis, which is limited to a few descriptive indications. The results should be expanded and detailed, discussed and reflected upon.

In our paper, we tried to maximally meet modern standards for the design of the results in accordance with our article. Further, the main objective of the study is to identify the effectiveness of methods of teaching a foreign language for professional purposes. The results of the study are reflected in charts, graphs and tables. We hope that we can tell you in more detail about each method used in our practice in the following articles. We are very grateful for your valuable comments.

The discussion also lacks consistency. There is no certain link to the conclusions. There are certainly no recommendations that exhort the results obtained.

In our paper, we study the teaching  foreign  professional language methods effectiveness. We could in some way identify the most effective methods using the survey. However, we cannot make recommendations, since this is beyond the scope of our competence.

Our research can be a guide for the teacher, but he makes the decision himself what method to use in teaching foreign languages for professional purposes.

We are very grateful to you.

With hope for further collaboration 

Best regards

Reviewer 2 Report

In general it is a good paper, well documented.

Author Response

Dear college,

We are very grateful  for your positive evaluation of our manuscript  and for review. 

With respect

Round 2

Reviewer 1 Report

The theoretical framework is significantly improved but statistical analysis is still very limited. I leave it up to the editorial board to decide on its publication, but this section should be improved. 

Author Response

Dear colleague!

First of all, we want to thank you for your precious attention and gigantic work paid to our work.

It is a great honor for us to work with a specialist of such high level in our field. Teaching foreign languages for professional purposes is a very difficult and delicate area.

We are very happy with our  common cooperation.

Thanks to your invaluable comments, we managed to improve  significantly our article, in our opinion.

As you recommended to us, we revised our tasks in the study, expanded and enriched the statistical analysis. We have added new figures to the study in order to best show the results of the work.

All changes are highlighted in yellow.

With hope for cooperation and a positive assessment from your side, we are waiting for your comments.

With great respect and best regards,

Authors